# Vascular Access in Pediatric Oncology and Hematology: State of the Art

**DOI:** 10.3390/children9010070

**Published:** 2022-01-05

**Authors:** Alessandro Crocoli, Cristina Martucci, Giorgio Persano, Maria Debora De Pasquale, Annalisa Serra, Antonella Accinni, Ivan Pietro Aloi, Arianna Bertocchini, Simone Frediani, Silvia Madafferi, Valerio Pardi, Alessandro Inserra

**Affiliations:** 1General Surgery Department, Bambino Gesu Children’s Hospital, IRCCS, 00165 Rome, Italy; alessandro.crocoli@opbg.net (A.C.); giorgio.persano@opbg.net (G.P.); antonella.accinni@opbg.net (A.A.); ivanpietro.aloi@opbg.net (I.P.A.); arianna.bertocchini@opbg.net (A.B.); simone.frediani@opbg.net (S.F.); silvia.madafferi@opbg.net (S.M.); valerio.pardi@opbg.net (V.P.); alessandro.inserra@opbg.net (A.I.); 2Paediatric Haematology/Oncology Cell and Gene Therapy Department, Bambino Gesu Children’s Hospital, IRCCS, 00165 Rome, Italy; mdebora.depasquale@opbg.net (M.D.D.P.); annalisa.serra@opbg.net (A.S.)

**Keywords:** vascular access, children, cancer

## Abstract

Management and successful use of vascular access are critical issues in pediatric patients affected by malignancies. Prolonged course of disease, complex and various treatment protocols require long-lasting vascular access providing adequate tools to administrate those therapies and to collect routine blood sampling without painful and repeated venipuncture. For these reasons, central venous catheters are currently an important component in pediatric onco-hematological care, with a direct influence on outcome. Indeed, there are peculiar issues (techniques of insertion, management, complications etc.) which must be well-known in order to improve the outcome and the quality of life of children with cancer.

## 1. Introduction

Although cancer is a rare condition in children, it represents the second most common cause of death in patients older than 12 months [1,2]. In 2017, National Institute of Health-National Cancer Institute estimates that there will be 10,270 new diagnoses of malignancies among patients from 0 to 14 years of age, and 1.190 children are expected to die from the disease [1]. Over the last five decades, the overall survival for this population has noticeably raised, with 5-years survival rate after diagnosis increased from 50% in late 70’s to 84% in 2020 [3]. Furthermore, a more remarkable improvement has been reported for tumors like acute lymphoblastic leukemia (5% in early 50’s to 91% in 2020) [4,5], lymphomas (30% in 60’s to around 98% in 2020) [6] and Wilms tumor (20% to 93%) [7,8]. All those successes have been accomplished since the introduction of the concept of multidisciplinary treatment in pediatric onco-hematology: more intensive chemotherapy and radiotherapy regimens, possibility of stem cells/bone marrow transplantation, introduction of new therapies (like immune and target therapy) and different surgical approaches have been developed over the last decades, playing an important role in different treatment protocols and having a direct influence on the outcome. However, important side effects of therapies (severe pancytopenia, chemotherapy/radiotherapy-induced mucositis, graft-versus-host disease etc.) frequently require more intense (and often challenging) infusion supportive treatments (e.g., blood component transfusions, apheresis, prolonged parenteral nutrition). Moreover, the necessity to have a safe and long-term way to obtain routine blood samples in pediatric patients, who are less prone to repeated and painful venipuncture (because of needle-phobia, reduced pain tolerance, coagulation disorders and risk of progressive peripheral vein depletion), represents another crucial factor determining the progressive popularity of central venous catheters in the modern management of children with onco-hematological diseases.

In 1973 Broviac et al. [9] developed and described the first long-term silicone central venous catheter to deliver home parenteral nutrition in patients, while its first application for onco-hematological therapies was reported in 1979 by Hickman et al. [10]. Gyves and co-workers then proposed the use of a totally implanted device for patients with cancer in 1982 [11,12]. In the last decades, many advances have been reported in terms of materials, techniques and care of patients with VADs; however, despite that many protocols and guidelines have been created, there is far less evidence for children than for adults, especially those of the onco-hematological variety.

The aim of this paper is the provide an updated review of the general recommendations for VADs management. 

## 2. General Principles

Patient’s requirements are the first and most important issue to consider for the choice of vascular access device (VAD), taking also into account expected duration, type and previous history of onco-hematology treatment. There is a large and established consensus that chemotherapy medication should be delivered through an adequate central venous access, in order to reduce to lowest terms the risk of infusion-related injuries. This statement must be applied also for supportive care issues, as well as for management of advanced cancer stages with palliative measures [13].

One of the main concerns for specialists treating onco-hematological children is the high volume and number of medications (either drugs or blood components) to administrate via the catheter. As a consequence, the erroneous idea of “the bigger lumen, the better performance” and “more lumens, easier management” lead to demand for large bore and/or multiple lumen catheter for all patients. 

Regarding catheter’s caliber, it has also been demonstrated that bigger catheters present higher risk of venous thrombosis; moreover, an outsized catheter requires venotomy during the placement procedure. For the above-mentioned reasons, the size of the device should be based on the diameter of the target vein measured with ultrasound, especially in cancer patients, who are more prone to coagulation disorders (both bleeding and hypercoagulation, depending on the phase of disease and the treatment they undergo) [14,15,16,17,18]. Preferably, the outer diameter of the catheter should not be higher than one-third of the diameter of the target vein chosen for placement: for example, a 3 Fr/1 mm catheter is appropriate for a vein whose diameter is 9 Fr/3 mm or larger.

Instead, multiple lumen catheters are indicated in patients undergoing intensive treatments or stem cell transplantation, considering the risk of undesirable reactions due to drug interacting during administration of multiple intravenous medications and solutions (e.g., drugs and inappropriate IV diluents, drug-drug incompatibility) [19]. For those reasons, dual lumen VADs are often recommended in these patients; however, management of multiple lumen catheters may link to higher risk of infection compared to single-lumen ones, so consequently their use as first-choice device is still debated [20,21,22,23,24] Furthermore, during the first 90 days of VADs, CLABSI (central line–associated bloodstream infections) incidence seem to be higher for single-lumen catheters (4.73/1.000 days for single-lumen VADs versus 1.54/1.000 days for double-lumen ones), whereas after 90 days the CLABSI incidence rates are higher in multiple-lumen catheters (2.44/1.000 days for double-lumen catheters versus 0.97/1.000 days for single-lumen ones) [25].

## 3. Indications

As a general rule, short-term catheters should be chosen in emergency settings (resuscitation, lifesaving apheresis procedures for hyperleukocytic leukemia/acute graft versus host disease treatment, acute renal failure etc.) and only for intrahospital use. Thus, most children require long-term venous access for discontinuous and intra/extrahospital use [26].

Toddlers/younger children, who are less tolerant to repeated and painful punctures, and patients undergoing intensive and prolonged chemotherapy regimens could benefit from a partially implanted/tunneled VAD (PI-TVAD), which is easy to handle and allows for repeated infusion/blood sampling without pain. Among their disadvantages are reported the following: disturbed body images, need for exit-site caring with regular dressing, limitations on physical activities (shower, sea/pool swimming etc.). Adolescents/young adults or children with discontinuous therapies (3–6 weeks interval) are eligible for totally implanted devices (TID) such as port catheters. These devices do not necessitate local care and do not limit any physical activities, preserving patients’ body image; however, their access (through skin puncture to the reservoir) may be more painful or difficult (in obese patients) and their removal necessitates of surgical procedure. Therefore, TID are recommended in patients requiring intermittent and prolonged use while PI-TVAD are indicated for prolonged but continuous/frequent vascular access [14]. 

It is nowadays well-known that peripherally inserted central catheters (PICC) represent a versatile, durable and accessible long-term vascular access in pediatric population, due to improvements in materials (III/IV generation power injectable polyurethane), quick placement procedure (bedside under local anesthesia) and good patients’ compliance. In children with cancer, PICCs are a good solution for patients who necessitate adequate vascular access and could not undergo to general anesthesia (for example in case of large mediastinal mass with airway compression/dislocation) as well as for patients who do not stand PI-TVAD coming out for the chest [27,28,29]. Their use has also been internationally described for patients undergoing bone marrow transplantation [30,31,32]. For these reasons, PICCs must be considered effectively as long-term VADs [16]. Furthermore, World Congress Vascular Access (WoCoVa) in 2013 proposed new VAD nomenclature and classification, no longer based on device “duration” but on vein incannulated during insertion: Centrally Inserted Central Catheters (CICCs) are devices inserted through a vein in the upper part of the body, particularly in the supraclavicular/subclavicular region, PICCs are those inserted through a deep vein of brachial region (basilic or cephalic vein), while Femorally Inserted Central Catheters (FICC) are placed with femoral vein puncture [33].

## 4. Placement Technique

Over recent years, evidences discouraging open venous cut-down (OSC) as primary technique for VADs placement have been reported, especially in children with cancer. This procedure should be considered a historical “legacy” of pediatric surgeons of last century [9,34,35]. Even if some authors still recommend OSC especially in children with coagulation disorders [36], vein surgical preparation and cannulation for vascular access is nowadays obsolete and then contraindicated especially for onco-hematological patients [13,14,16,33,37]. Indeed, venous OSC is related with higher risk of hemorrhage (for wider tissue dissection during vein preparation, incision of vein wall with subsequent suture etc.), as well as early dislodgment and infection (secondary to surgical procedure) [38,39] and its use should be re-served to exceptional cases. Moreover, OSC require skills and knowledge of vascular and micro-surgery and, in the case of catheter failure (both for end of treatment and for complications), the possibility of vein permanent stenosis/thrombosis has been reported.

Recently minimally invasive procedures for catheter placement have become the gold standard in adult patients. Though the evidence in pediatric patients is still debated, most studies suggest that ultrasound guidance should become the main technique of venipuncture also in children and neonates [16,33,40]. Its most important advantage is the possibility to choose, before or during the procedure, the most appropriate vein after a scan of all possible options. Furthermore, via ultrasound guidance, puncture and cannulation of the vein are quicker and easier, as well as associated with fewer complications (such as infection of the insertion site, peri-procedural bleeding and catheter-related thrombosis), due to the reduced invasiveness of the procedure [37,41].

Recommendations from AIEOP for VADs positions are reported in Table 1. 

## 5. Prevention and Management of Complications

Considering the prolonged course of the disease and the significative improvement in survival rate for children with cancer, strategies to prevent early and long-term complications and to preserve vascular assets directly influence outcome and quality of life for these patients. It is mandatory to acknowledge all the possible scenarios, to adopt preventive strategies and overcome eventual difficulties through prompt recognition and appropriate management.

As previously mentioned, minimally invasive ultrasound-guided procedures for VADs placement totally changed the approach to vascular access, especially in pediatric patients. With sufficient training and knowledge of devices available (needle, wires, introducers, catheters, probes etc.) and technique, ultrasound consents high percentage of first-time success and allows diagnosis of preexisting vascular anomalies (malformations, anatomic variants or thrombosis/stenosis), reducing the incidence of periprocedural complications (arterial inadvertent puncture, hematoma, extravasation, dissection of the vessel or stenosis etc.) [16]. Moreover, avoiding multiple punctures reduces hemorrhagic risk in these patients who often have coagulation disorders after chemotherapy (e.g., thrombocytopenia, liver failure in veno-occlusive disease). Use of ultrasound has also been reported for early diagnosis of pneumothorax, which represents a common complication in subclavian vein cannulation with blind technique [42,43].

Catheter dislodgment and/or tip migration may lead to malfunction of the device (either in infusion or in blood draw) and, in worst cases, to complete removal. Pediatric patients undergoing chemotherapy and/or high dose steroids are more prone to these complications for various reasons, such as reduced healing ability and coagulation, increased infection, skin and subcutaneous tissue disease (like graft-versus-host disease). Different approaches to reduce these events have been described, such as the use of non-cuffed third generation polyurethane (which are also less subject to rupture compared to silicone-made devices), secured with both suture-less devices and subcutaneously anchored securement systems (SASS) [14,44,45]. Use of cyanoacrylate glue on the exit-site has also been described as a valid option to reduce failure of VADs in onco-hematological children (due to better securement, reduced infection rate for quick skin healing and valid hemostatic action) [46,47,48].

Recommendations from AIEOP for VADs management are reported in Table 2.

## 6. Catheter Removal

Catheter removal is widely considered a neglected procedure, frequently inaccurately performed by skilled staff members in different environments (bedside under local anesthesia, in a procedure room with mild sedation etc.), according to personal or institutional experience, and not always following evidence-based protocols. Nevertheless, in the case of patients with long-term cuffed devices (e.g., Hickman/Broviac/Leonard), catheter removal represents a “real” surgical procedure with non-negligible risk of complications (hemorrhage, fracture of catheter with hembolization, etc.) [49,50]. For children with cancer, catheter removal must also be considered as one of the many painful procedures they undergo during the course of disease, with additional stress for the patients and their families. For these reasons, and in case of elective long-term cuffed catheter removal (for example, at the end of therapy purposes), the procedure should be done in a dedicated room equipped with adequate facilities and under sedation, with trained medical staff and according to institutional guidelines [51]. Surgical maneuvers must include blunt dissection and removal of the polyester cuff, together with the device; retained cuff may represent an innocuous foreign body, but, in some cases, may determine infection/chronic inflammation, false image (calcification deposits or metastases) at radiology and unaesthetically reaction on the skin [50,51,52]. As a matter of fact, introduction of new devices with different securement systems (suture-less de-vices, cyanoacrylate glue and SASS) lead both to easier fixation and removal of the catheter if necessary, eliminating the issue of polyesther cuff-equipped catheters, whose adoption should be progressively abandoned in pediatric patients with cancer.

## 7. Conclusions

In recent years, many improvements in terms of materials, techniques and care have been described; however, despite many protocols and guidelines being reported over the last decades, there is far less evidence for pediatric patients than for adults, especially onco-hematological ones. Appropriate knowledge and skilled teams are mandatory in order to address issues regarding indications, placement and removal techniques, choice of the devices, catheter care and possible complications in such a peculiar population.

## Figures and Tables

**Table 1 children-09-00070-t001:** AIEOP Recommendations for VADs Positioning *.

(1) A tunneled catheter is recommended for continuous use (A I) (2) For discontinuous use, a totally implanted VAD is recommended (A II) (3) It is recommended that the ratio of the catheter caliber to vein diameter should not exceed 1/3 (A II) (4) Multiple lumen VADs should be inserted only in few selected patients, based on the intensity of care and on the therapeutic program (A) (5) The choice of material must be based on high performance in terms of guaranteed flows and pressure resistance as well as device endurance (A II) (6) Insertion by surgical venous cutdown is not recommended (A I) (7) The ultrasound-guided technique represents the current standard for venipuncture and venous cannulation for insertion of VAD (A It) (8) The use of cyanoacrylate tissue glue is recommended (A II)

* The grading of evidence based medicine is reported according to the European Society of Microbiology and Infectious Disease. VAD—vascular access device (VAD), AIEOP—the Italian Association of Pediatric Hematology and Oncology.

**Table 2 children-09-00070-t002:** AIEOP Recommendations for VADs Management *.

(A) Minimize the number of VADs accesses to prevent infections, doing diagnostic and therapeutic procedures at the same time (A) (B) Limit intermittent infusion (A II) (C) Use NFC (needle free connectors) (A II t) (D) Respect the aseptic technique in the management of VADs and during dressings, changes, using only sterile, single use devices (A r) (E) Minimize the number of additional devices (ramps, filters, caps, extensions) to reduce the risk of contamination and accidental disconnections (A It) (F) Carefully examine the catheter exit site and the sur- rounding area daily (without removing the dressing if not necessary) to identify any redness, tender- ness, edema, and secretions (A II) (G) Use 2% chlorhexidine gluconate in 70% isopropyl alcohol as a skin antiseptic to clean the exit site. Single-dose preparations of chlorhexidine reduce the risk of microbial contamination (A It)

* The grading of evidence based medicine is reported according to the European Society of Microbiology and Infectious Disease. VAD—vascular access device (VAD), AIEOP—the Italian Association of Pediatric Hematology and Oncology.

## Data Availability

No new data were created or analyzed in this study. Data sharing is not applicable to this article.

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
