# Peer review of "Vascular Access in Pediatric Oncology and Hematology: State of the Art"

_children, 2022, doi:10.3390/children9010070_

Round 1
Reviewer 1 Report
I thank the Authors for the submission of this paper. I think it is an interesting and still little researched topic, and the work is original. At the same time, it is very important to produce strong evidences about the use, management and care of vascular access in children, particularly for those with oncological disease.
There are several issues that should be considerd in a more comprehensive and clear way. I have included a small list of suggestions for improvement which I hope may be helpful to you.
Despite being a very accurate review of the state of the art on this topic, in my opinion the aim of the work is missing. It should be well declared, in order also to information on the usefulness of the work for the scientific community. Therefore, please define clearly the aim of this paper and add a conclusive paragraph.
Is it possible, for some of the issues explored, to make a quantitative synthesis of the evidence found?
Lines 53-56: I think that this part could be more suitable for a conclusion paragraph
The 4th point of the paper should be 'Placement techniques'.
Please revise the English form: in some parts of the text it seems that words or verbs are missing, and minor typing errors could be found (e.g. "pro-longed", "total-ly", "de-vices").
Author Response
I thank the Authors for the submission of this paper. I think it is an interesting and still little researched topic, and the work is original. At the same time, it is very important to produce strong evidences about the use, management and care of vascular access in children, particularly for those with oncological disease. There are several issues that should be considered in a more comprehensive and clear way. I have included a small list of suggestions for improvement which I hope may be helpful to you.
Thank you very much for your review. We found your suggestions very useful to improve our manuscript.
Despite being a very accurate review of the state of the art on this topic, in my opinion the aim of the work is missing. It should be well declared, in order also to information on the usefulness of the work for the scientific community. Therefore, please define clearly the aim of this paper and add a conclusive paragraph. Lines 53-56: I think that this part could be more suitable for a conclusion paragraph
We created a “Conclusion” paragraph, according to your suggestion, moving the mentioned lines, and defined clearly the aim of our paper in the Introduction.
Is it possible, for some of the issues explored, to make a quantitative synthesis of the evidence found?
We add two summarizing tables in the manuscript to provide the most recent evidences.
The 4th point of the paper should be 'Placement techniques'.
We changed the title of this paragraph, according to your suggestion.
Please revise the English form: in some parts of the text it seems that words or verbs are missing, and minor typing errors could be found (e.g. "pro-longed", "total-ly", "de-vices").
We made a language revision and corrected the typing errors.
Reviewer 2 Report
Review; state of the art central venous catheters
Thank you for your manuscript. Few comments, see below
Vascular Access in Pediatric Oncology and Hematology: State of the Art
What is new if you compare it to your paper 2015 Central venous access devices in pediatric malignancies: a position paper of Italian Association of Pediatric Hematology and Oncology in Journal of vascular access. That manuscript positions very well with the available evidence when which catheter should be inserted and what precautions should be taken. Please comment
In your introduction you update the survival rates to 2010. As the manuscript is in 2021 I would recommend updating this to 2020 for all the malignancies stated with the available literature.
All those successes have been accomplished since the introduction of the concept of mul- 32 tidisciplinary treatment in pediatric onco-hematology. You should mention newer therapies here as well with immune therapy and targeted therapy
Moreover, the necessity to have a safe and long-term way to obtain routine blood samples in pediatric patients,I would be careful mentioning this. Many guidelines are careful with advising this as you open the catheter and can introduce infection. Less handling of the catheter of the catheter is required to guarantee optimal use.
For those reasons, dual lumen VADs are often recommended in these patients; however manage ment of multiple lumen catheters may link to higher risk of infection compared to single lumen ones, consequently their use as first-choice device is still debated I would add risks of CRBSI per 1000 days for the different catheters so one gets a feeling of what this means for the patient
Please comment on all items above in your introduction
Indications; Toddlers/younger children, who are less prone to repeated and painful punctures, What do you mean by this I think you mean Toddlers are even less tolerant to painful procedures, adjust
Therefore, TID are recom- mended in patients requiring intermittent and prolonged use while PI-TVAD are indi cated for prolonged but continuous/frequent vascular access Again that into account the risk of infection. In TID risk of infection is lower therefore the preferred catheter to insert
Please comment
Picc catheters can be a good solution in certain circumstances but also here add the risk for infection and thrombosis. Please add the risks and reformulate
Prevention and management of complications
I disagree with you on placing non cuffed catheters in children on high dose steroids
All hematology oncology patients are on longterm steroids and need a catheter for about 2 years
So they need a cuffed catheter Please comment
Catheter removal
As a matter of fact, introduction of new devices with 193 different securement systems (suture-less de-vices, cyanoacrylate glue and SASS) lead to both easier fixation and removal of the catheter if necessary, eliminating the issue of pol yesther cuff equipped catheters, whose adoption should be progressively abandoned in pediatric patients with cancer. I hope you do not mean that we are not placing cuffed catheters in future. All our kids need catheters between 6 months and 2-3 years which makes the long term cuffed tunneled central venous catheter the only option
Author Response
What is new if you compare it to your paper 2015 Central venous access devices in pediatric malignancies: a position paper of Italian Association of Pediatric Hematology and Oncology in Journal of vascular access. That manuscript positions very well with the available evidence when which catheter should be inserted and what precautions should be taken. Please comment
Comparing with 2015 paper, we introduce some new highlights: bundle of care, use of SADs, specific guidelines for catheter removal. We also adopted the new nomenclature by WoCoVA (CICC/FICC etc).
In your introduction you update the survival rates to 2010. As the manuscript is in 2021 I would recommend updating this to 2020 for all the malignancies stated with the available literature.
We updated the survival rate to 2020; statistics are adapted from the American Cancer Society and National Institutes of Health MedlinePlus websites.
All those successes have been accomplished since the introduction of the concept of multidisciplinary treatment in pediatric onco-hematology. You should mention newer therapies here as well with immune therapy and targeted therapy
We mentioned immune and target therapy in the sub mentioned paragraph.
Moreover, the necessity to have a safe and long-term way to obtain routine blood samples in pediatric patients. I would be careful mentioning this. Many guidelines are careful with advising this as you open the catheter and can introduce infection. Less handling of the catheter of the catheter is required to guarantee optimal use.
Despite many guidelines advice against the use of VADs to obtain blood samples in order to reduce infections, in our institution, we frequently get blood drawn from VADs in order to preserve the venous heritage (which is already poorer in pediatric patients than adults) and to reduce painful procedures in this population, especially in in-patient settings or prolonged hospitalizations.
For those reasons, dual lumen VADs are often recommended in these patients; however management of multiple lumen catheters may link to higher risk of infection compared to single lumen ones, consequently their use as first-choice device is still debated I would add risks of CRBSI per 1000 days for the different catheters so one gets a feeling of what this means for the patient.
We agree with this evaluation; we frequently debate with our onco-hematologists for the use of double vs single lumen catheter, for the higher risk of complication related to dual lumen devices. We added the requested information and the relative reference.
Indications; Toddlers/younger children, who are less prone to repeated and painful punctures, What do you mean by this I think you mean Toddlers are even less tolerant to painful procedures, adjust
We agree with your comment and corrected this sentence.
Therefore, TID are recommended in patients requiring intermittent and prolonged use while PI-TVAD are indicated for prolonged but continuous/frequent vascular access Again that into account the risk of infection. In TID risk of infection is lower therefore the preferred catheter to insert. Picc catheters can be a good solution in certain circumstances but also here add the risk for infection and thrombosis. Please add the risks and reformulate
PICCs are equally long-term devices, but they are indicated in specific population (older patients, without renal disease etc). For these catheters, the risk of thrombosis is, as well, correlated with the ratio between the diameter of the catheter and the diameter of the vein and it is not higher than the one reported with other devices. Furthermore, some authors demonstrated that the risk of infection may be reduced if the exit-site is far from the venipuncture (Dawson, Robert. (2011). PICC Zone Insertion Method™ (ZIM™): A systematic approach to determine the ideal insertion site for PICCs in the upper arm. Journal of the Association for Vascular Access. 16. 156-165. 10.2309/java.16-3-5.).
I disagree with you on placing non cuffed catheters in children on high dose steroids. All hematology oncology patients are on long-term steroids and need a catheter for about 2 years. So they need a cuffed catheter. As a matter of fact, introduction of new devices with 193 different securement systems (suture-less devices, cyanoacrylate glue and SASS) lead to both easier fixation and removal of the catheter if necessary, eliminating the issue of polyesther cuff equipped catheters, whose adoption should be progressively abandoned in pediatric patients with cancer. I hope you do not mean that we are not placing cuffed catheters in future. All our kids need catheters between 6 months and 2-3 years which makes the long term cuffed tunneled central venous catheter the only option
I agree with your skepticism about the use of non-cuffed catheters, especially for long-term use, but the current application of subcutaneously anchored securements allows to reduce significantly the risk of dislodgement, as well as local pain or inflammation. So, in specific population (like the ones undergoing high dose steroids), these devices may permit the use of non-cuffed catheters.
Round 2
Reviewer 2 Report
I do believe the manuscript has improved by answering to the reviewers